# A Video Captioning Method Based on Multi-Representation Switching for Sustainable Computing

Heechan Kim [1] and Soowon Lee [2,*]

1   Department of Software Convergence, Soongsil University, Seoul 06978, Korea; heathkim@soongsil.ac.kr
2   School of Software, Soongsil University, Seoul 06978, Korea
*   Correspondence: swlee@ssu.ac.kr

**Abstract:** Video captioning is a problem that generates a natural language sentence as a video's description. A video description includes not only words that express the objects in the video but also words that express the relationships between the objects, or grammatically necessary words. To reflect this characteristic explicitly using a deep learning model, we propose a multi-representation switching method. The proposed method consists of three components: entity extraction, motion extraction, and textual feature extraction. The proposed multi-representation switching method makes it possible for the three components to extract important information for a given video and description pair efficiently. In experiments conducted on the Microsoft Research Video Description dataset, the proposed method recorded scores that exceeded the performance of most existing video captioning methods. This result was achieved without any preprocessing based on computer vision and natural language processing, nor any additional loss function. Consequently, the proposed method has a high generality that can be extended to various domains in terms of sustainable computing.

**Keywords:** video captioning; multi-representation switching; deep learning





## 1. Introduction

With the remarkable development of big data techniques and computing power based on graphic processing units, deep learning research developed from neural networks is being actively conducted. In domains such as computer vision and natural language processing, there are many studies on deep learning methods that outperform classical machine learning methods.

Video description is a problem in which natural language sentences are generated for a given video that consists of continuous images. This is a research field that integrates research on image processing and natural language processing. This problem can be seen as a problem of summarizing a video. In the past, research on computer vision, such as image classification, and research on natural language processing, such as automatic summarization, have been conducted separately. In recent years, owing to the increasing demand for multi-modal data analysis, such as automatic video subtitling and video surveillance, many studies that integrate computer vision and natural language processing are being actively conducted. Video description can be subdivided into three problems according to the density of the description for a given video. One is the video captioning problem, which creates a short description for a given video. The second is the dense video captioning problem, which explains a situation according to the temporal events in the video. The third is the video question answering problem, which answers a given video and natural language query [1].

In this study, we focus on the video captioning problem of generating an overview description for short video clips, which are seconds or tens of seconds in length. Video captioning is a more difficult problem than image captioning because it is necessary to extract information on objects and their properties and behaviors that exist in multiple

frames in the video, not a single image. To solve this problem, various methods have been proposed to learn the sequential structure of videos [2–21].

Most existing methods have focused on the method of extracting representations, including the characteristics of entities and motions in the video. However, these methods have a problem, that is, it is difficult to ascertain whether a given word represents video information or is necessary for a grammatical reason. For example, if there is a video with the description "a man is walking with a dog," the words "man," "walking," "dog," and "with" represent the entities or the relationships among the entities in the video, while the words "a" and "is" are needed for grammatical reasons.

Currently, many studies in the fields of computer vision and natural language processing solve problems using methods with very high computational complexity. These methods require a large amount of resources for training and testing. In the application stage, this can be a big problem for tasks that require a fast response. Thus, for such sustainable computing, studies have been proposed to develop methods that show similar or better performance but consume fewer resources [22–25]. From this observation, we present two research questions. One is that whether a complex feature extraction process from a video is essential to generate proper words. The question is whether a method with a simpler feature extraction structure can achieve similar or better performance. The other is whether it is possible to model sentence structure according to grammar using next word prediction settings, which is widely used in the field of natural language processing. This question is to check whether a method widely used in the field of natural language processing can achieve good performance in multi-modal data with minimal modifications.

To answer both questions, we set up the following hypotheses: First, the characteristics of a video can be classified into an entity characteristic in each frame of the video and a motion characteristic in multiple frames of the video. Second, the characteristics of entity and motion can be captured in separate steps. Third, sentence structure can be learned effectively when the information in the video and the text are handled separately.

To verify these hypotheses, we propose a multi-representation switching method for video captioning that selects either a visual representation or a textual representation to generate the next word. The proposed method is a deep learning model with a compact structure, which consists of three components for learning the characteristics of input data separately. The main contributions of this research are as follows:

- We present a deep learning method that can be trained in an end-to-end manner and does not require any other preprocessing based on computer vision, natural language processing, or any additional loss function.
- The proposed method can be used as a framework for various tasks that are provided with sequential data because the method consists of compact feature extraction phases.

We verify the research questions and the following hypotheses through experiments. Because the proposed method rarely requires prior domain knowledge, it has high generality that can be easily applied to various problems. In this respect, we use the proposed method to present a new example from the perspective of sustainable computing.

The remainder of this paper is organized as follows. Related works are described in Section 2. The proposed method is described in Section 3, and its performance evaluation is presented in Section 4. The conclusion and future work are explained in the last section.

## 2. Related Works

In the field of computer vision, there are studies that accurately classify objects in images better than humans [26,27]. In addition, a method of extracting personal characteristics by recognizing a person's face and focusing on individual information has been studied [28,29]. In the past, it was difficult to model an image when several entities were mixed in the image. To alleviate this problem, a method of modeling the relationship between entities in an image as a graph was studied [30]. In the medical field, an image generation method was studied to solve the problem of being highly overfitted with patient information [31].

In the natural language processing of automatic summarization, the loss of input information should be considered. To consider this, a method of modeling the input document as the entire context rather than a word has been studied [32]. To directly model the loss of information itself, a method of modeling the previously output information as adaptive noise was studied [33].

Video captioning tasks can be divided into a template-based approach and an approach based on sequential classification learned in an end-to-end manner. Template-based methods generate a video description by slot filling appropriate words after selecting an output sentence structure from a set of predefined sentence structures according to the video [34–38]. Kojima et al. proposed a method that uses the concept hierarchy of actions [34]. To deal with the lack of textual features, Guadarrama et al. proposed a method using semantic relationships of phrases [35]. By contrast, Rohrbath et al. proposed a method that uses relationships between entities and motions that are modeled by conditional random fields and consistencies between words and their topics [36,37]. Xu et al. proposed a method that jointly embeds a video and a sentence into a common space by using a dependency tree structure for sentences [38].

Sequential classification-based methods generate a description of a video by predicting the next words recursively [2–21]. Venugopalan et al. proposed a method that couples a convolution neural network (CNN) and a sequence-to-sequence structure, which is widely used in natural language processing fields such as automatic summarization [2,3]. This method represents video information as a single vector, and then generates a sequence of words from this single vector. This method, therefore, has a problem in that the information of each frame cannot be selected dynamically according to the importance of the word to be decoded.

To alleviate this problem, Yao et al. proposed a method that applied a temporal attention mechanism [39] to evaluate the importance of each frame [4]. This method evaluates the importance of each frame based on the attention mechanism. Based on this importance, this method can aggregate video information dynamically when generating word probability distributions. Oriented gradients, oriented flow, and motion boundaries, which were separately extracted from the video, were used as inputs for the method.

The previous methods used only the final representation of CNNs, and there was a limitation in that the image representation according to the granularity of the image could not be considered. To model image representations for various granularities, Ballas et al. proposed a method that abstracts representations of each frame extracted from multiple CNN blocks using a recurrent neural network (RNN) [5]. To consider the important locations in frames, Yan et al. proposed a method that extracts video features using a two-dimensional CNN, a three-dimensional CNN [40], and regions with CNN features [41] and generates a description using spatio-temporal attention [6].

Existing methods have a structure that extends the approach used for images to videos. Video data can be seen as a set of small video clips captured from different viewpoints. To model the characteristics of a video, Baraldi et al. proposed a method that recognizes the segmentation of a video using a two-layered RNN encoder; one layer is for detecting the segmentation, and the other is for learning the entire video representation [7]. From the perspective of selecting only the salient frame, Chen et al. proposed a method that selects only the important frames without using an attention mechanism [8]. This method was trained by reinforcement learning based on rewards for the accuracy of generated descriptions and the diversity of selected frames. To extract the entity and its motion characteristics from a video, Wu et al. proposed a method that uses the trajectory grouping of the video and the dependency tree of words based on the trajectory information [9].

In addition to mapping the embedding of each frame to the embedding of words in a direction, Wang et al. proposed a method that leverages both the forward (video to sentence) and backward (sentence to video) flows [10].

There are also studies that approach the video captioning problem from the viewpoint of natural language processing. To hierarchically model the meaning of a sentence and

paragraph of a video description in a decoder, Yu et al. proposed a method that has a two-layered RNN decoder, one for learning the representation of sentences, and the other for learning the representation of paragraphs [11]. Yang et al. proposed a generative adversarial network model [42] for video captioning [12].

## 3. Proposed Method

### 3.1. Motivation and Architecture

Looking closely at the data for video captioning, a given video consists of a sequence of images, which are called frames, and a given description consists of a sequence of words. An example of the relationship between a video and a description can be expressed as shown in Figure 1.

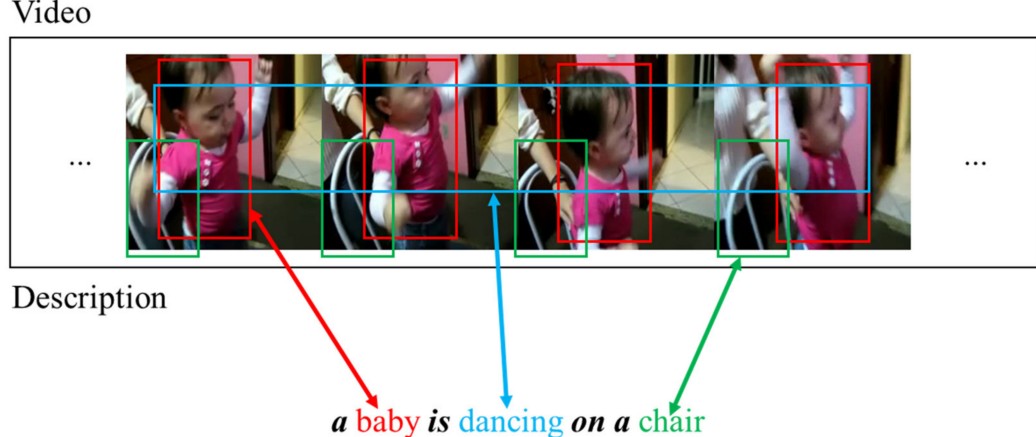

**Figure 1.** Example of the relationships between a video and a description.

The video in Figure 1 shows a baby dancing on a chair. In this example, the words "baby" and "chair" represent entities and the word "dancing" represents a motion in the video. These words are expressed in different colors. The words "a" and "is" are words necessary for grammatical reasons. The word "on" expresses the relationship between the entities. It can be seen that the entities and motion information in the video have direct relationships with the words in the description. However, the words that represent the relationship between entities and the words for the necessity of grammar are words that can only be expressed by learning the natural language.

To learn the relationship between this video and the description without a natural language processing method, such as part-of-speech tagging, the deep learning model should have a structure that can learn these two types of words separately. To achieve this, we propose a multi-representation switching method that allows the model to learn visual characteristics and textual characteristics. This method is based on the encoder–decoder framework, and its conceptual architecture is shown in Figure 2.

The left side of Figure 2 shows the encoder of the proposed method extracting features of the video using only a two-dimensional (2D) CNN and an RNN. The main idea behind this architecture is to construct a model with high generality based on the fact that a verb can be sufficiently modeled during image captioning using a single frame as an input. The 2D CNN extracts the entity features of each frame, and the RNN extracts the motion features using the sequence of these entity features.

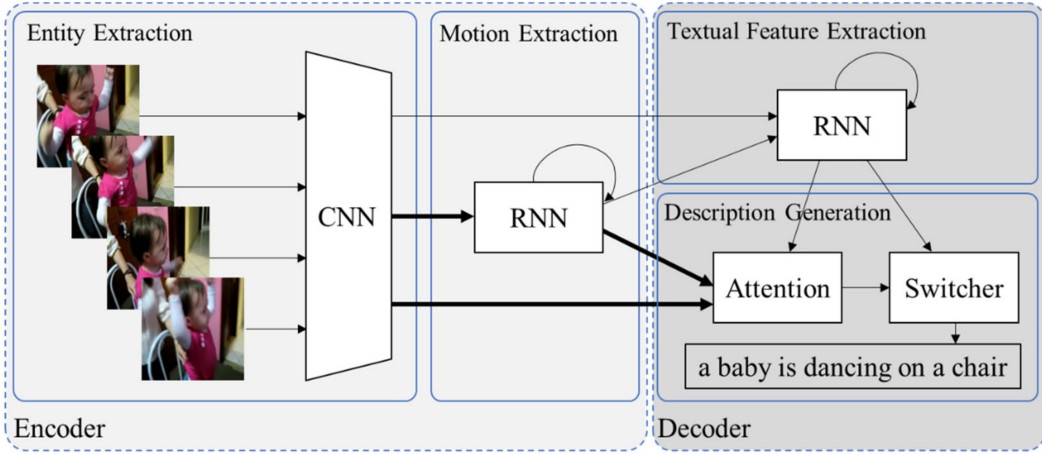

**Figure 2.** Conceptual architecture of the proposed method.

The right side of Figure 2 shows the decoder extracting a textual feature from the word sequence information using an RNN. The word sequence is generated using the entity and motion features extracted from the encoder and the textual features of the decoder. At this time, the entity and motion information required when generating each word are selected through the attention mechanism. A switcher generates the next word by selecting the extracted entity, motion, and textural features.

In Figure 2, a thin solid line indicates that one vector is transferred to the destination, and a thick solid line indicates that several vectors are transferred. For example, the thick solid line between the CNN and the RNN in the encoder implies that the vectors of each frame extracted by the CNN are transferred to the RNN.

### 3.2. Feature Extraction in the Encoder

The proposed method is based on an encoder–decoder framework to learn video and description pairs in an end-to-end manner. The encoder can be divided into two components. One is a CNN-based entity feature extraction component, while the other is an RNN-based motion feature extraction component. The inputs of the CNN are vectors of frames $\{\dot{v}_1, \ldots, \dot{v}_i, \ldots, \dot{v}_I\}$ of video $V$, where each frame, $\dot{v}_i$, is represented by a $224 \times 224 \times 3$ tensor. Each frame includes only RGB color information, and there is no additional information such as optical flow. Various CNN structures can be used, but in this study, we use the ResNet50V2 network [27], which shows good performance with relatively few parameters. We use the output of each stacked block of the ResNet50V2 network to extract object information of various granularities from the video. We select and use a block whose output size changes between the stacks of the convolution block of the ResNet50V2 network. Each frame is represented by six vectors, including the output vector of the final layer. We refer to these vectors as entity representations, and the graphical illustration is shown in Figure 3.

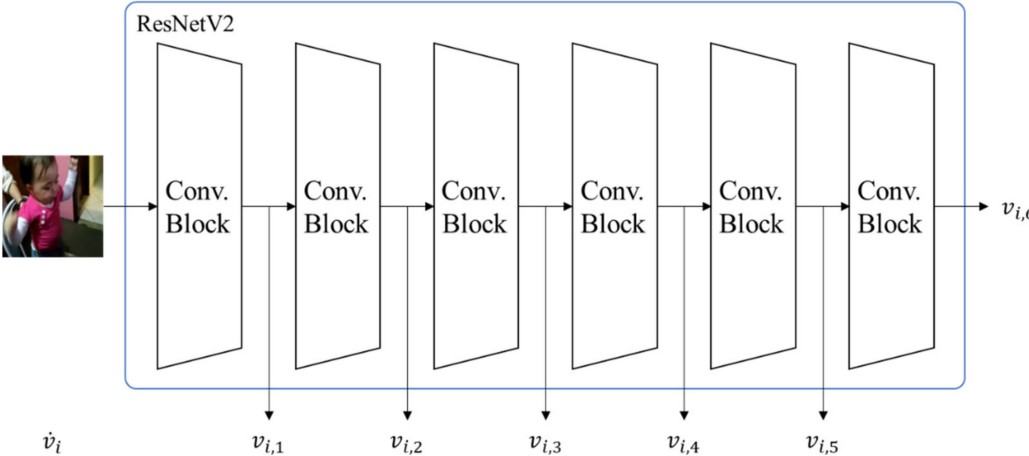

**Figure 3.** Graphical illustration of entity feature extraction from the ResNetV2 network.

We denote the ResNet50V2 network as a function $\varphi$, which is defined as follows:

$$\begin{bmatrix} \vdots \\ v_{ik} \\ \vdots \end{bmatrix} = \varphi(\dot{v}_i), \quad 1 \le k \le 6, \tag{1}$$

and the learnable parameters of the CNN are not expressed for readability. Details of each entity feature are provided in Table 1.

**Table 1.** Detail information of entity representations.

| Representation | Layer Name | Dimensionality |
|:---:|:---:|:---:|
| $v_{i1}$ | pool1_pool | $56 \times 56 \times 64$ |
| $v_{i2}$ | conv2_block3_out | $28 \times 28 \times 256$ |
| $v_{i3}$ | conv3_block4_out | $14 \times 14 \times 512$ |
| $v_{i4}$ | conv4_block6_out | $7 \times 7 \times 1024$ |
| $v_{i5}$ | conv5_block3_out | $7 \times 7 \times 2048$ |
| $v_{i6}$ | predictions | 1000 |

The layer names mentioned in Table 1 may vary depending on the deep learning library. To learn motion features using these entity representations, we apply a mean pooling operation to each of these representations and then embed them with the same dimension, $n_{ei}$. The final entity representation, $\widetilde{v}_i$, of the $i$-th frame by these operations is defined as follows:

$$\widetilde{v}_i = \begin{bmatrix} \cdots & E_k^{img}\tau(v_{ik}) & \cdots \end{bmatrix}, \quad 1 \le k \le 6, \tag{2}$$

and $E_k^{img} \in \mathbb{R}^{n_{ei} \times n_{kd}}$ is an embedding matrix for each entity representation, and the matrix is a learnable parameter. $n_{kd}$ represents the number of the last dimension of the $k$-th representation. For example, $n_{1d}$ is 64. $\tau$ represents the mean pooling operation. The final entity representation, $\widetilde{v}_i$, of the $i$-th frame is a $6n_{ei}$-dimensional vector.

The proposed method learns the motion feature of the video with an RNN based on the representation extracted by the CNN. In this study, we use a long short-term memory network (LSTM) [43] as the RNN and denote the LSTM in the encoder as a function $\psi^E$. The hidden state of the LSTM, $h_i^E$, in the $i$-th frame is defined as follows:

$$h_i^E = \psi^E\left(\widetilde{v}_i, h_{i-1}^E\right) \in \mathbb{R}^{n_{ri}}, \tag{3}$$

and the learnable parameters of the RNN are not expressed for readability. The initial hidden state, $h_0^E$, of the encoder's LSTM is defined as a zero vector. We refer to this vector as the motion representation.

### 3.3. Feature Extraction in the Decoder

The decoder of the proposed method consists of two components. One is a textual feature extraction component, and the other is a description generation component. Given a description that consists of a sequence of words, we use the LSTM to extract a textual feature. Similar to the encoder, the LSTM in the decoder is represented by a function $\psi^D$. The hidden state of the LSTM, $h_t^D$, in the $t$-th step is defined as follows:

$$h_t^D = \psi^D\left(E^w y_{t-1}, h_{t-1}^D, \overline{V}\right) \in \mathbb{R}^{n_{rw}}, \tag{4}$$

and the learnable parameters of the RNN are not expressed for readability. $y$ denotes a one-hot vector for the word. $E \in \mathbb{R}^{n_{ew} \times n_V}$ represents an embedding matrix; therefore, $E^w y_{t-1}$ represents an $n_{ew}$-dimensional embedding vector of the word $y$. $E^w$ is a learnable parameter. The initial hidden state, $h_0^D$, and cell state of the decoder's LSTM are defined by the last hidden state, $h_I^E$, and cell state of the encoder. $\overline{V}$ denotes a concatenated vector of mean vectors of entity representations, $\widetilde{v}_i$, and motion representations, $h_i^E$, in overall frames, and is defined as follows:

$$\overline{V} = \frac{1}{I}\left[\sum_{i=1}^{I}\widetilde{v}_i \sum_{i=1}^{I} h_i^E\right] \in \mathbb{R}^{n_{ei}+n_{ri}}. \tag{5}$$

### 3.4. Description Generation Using Multi-Representation Switching

Because the LSTM of the decoder is provided only with the entire video information and previous word information, there is a need to refer to information in important frames when generating words. For this reason, we use the attention mechanism [39] to aggregate proper video information for decoding. This aggregated information is represented as a context. The entity context, $c_t^e$, for the $t$-th step is the weighted sum of the entity representations, $\widetilde{v}_k$, which is defined as follows:

$$c_t^e = \sum_{k=1}^{I} \alpha_{kt}^e \widetilde{v}_k \in \mathbb{R}^{n_{ei}}, \tag{6}$$

and $\alpha_{kt}^e$ represents the attention scores between the $k$-th entity representation, $\widetilde{v}_k$, and the $t$-th textual representation, $h_t^D$, and it is defined as follows:

$$\alpha_{it}^e = \varsigma\left(w_e^\top \tanh\left(W_e \widetilde{v}_i + W_e' h_t^D\right)\right) \in \mathbb{R}, \qquad 0 \leq \alpha_{it}^e \leq 1, \qquad \sum_{k=1}^{I}\alpha_{kt}^e = 1, \tag{7}$$

and $\varsigma$ represents a softmax function. $W_e \in \mathbb{R}^{n_a \times n_{ri}}$, $W_e' \in \mathbb{R}^{n_a \times n_{rw}}$, and $w_e \in \mathbb{R}^{n_a}$ are learnable parameters.

Likewise, the motion context, $c_t^m$, for the $t$-th step is the weighted sum of the motion representations, $h_i^E$, which is defined as follows:

$$c_t^m = \sum_{k=1}^{I} \alpha_{kt}^m h_i^E \in \mathbb{R}^{n_{ri}}, \tag{8}$$

and $\alpha_{kt}^m$ represents the attention scores between the $k$-th motion representation, $h_i^E$, and the $t$-th textual representation, $h_t^D$, and it is defined as follows:

$$\alpha_{it}^m = \varsigma(w_m^\top \tanh(W_m h_i^E + W_m' h_t^D)) \in \mathbb{R}, \qquad 0 \leq \alpha_{it}^m \leq 1, \qquad \sum_{k=1}^{I}\alpha_{kt}^m = 1, \tag{9}$$

and $W_m \in \mathbb{R}^{n_a \times n_{ri}}$, $W'_m \in \mathbb{R}^{n_a \times n_{rw}}$, and $w_m \in \mathbb{R}^{n_a}$ are learnable parameters.

Through the above process, we can obtain the entity and motion information of the video and the previous textual information. In the $t$-th step, word logits, $P_t^e$, $P_t^m$, and $P_t^T$, for the next word based on each fixed size representation, $c_t^e$, $c_t^m$, and $h_t^D$, are defined as follows:

$$
\begin{aligned}
cP_t^e &= \sigma(W_P^e c_t^e + b^e) \in \mathbb{R}^{n_P}, \\
P_t^m &= \sigma(W_P^m c_t^m + b^m) \in \mathbb{R}^{n_P}, \\
P_t^T &= \sigma(W_P^T h_t^D + b^T) \in \mathbb{R}^{n_P},
\end{aligned}
\tag{10}
$$

and $\sigma$ represents an activation function. $W_P^e \in \mathbb{R}^{n_P \times n_{ei}}$, $W_P^m \in \mathbb{R}^{n_P \times n_{ri}}$, $W_P^T \in \mathbb{R}^{n_P \times n_{rw}}$, and $b^e, b^m, b^T \in \mathbb{R}^{n_P}$ are learnable parameters.

When generating the next word, we propose a multi-representation switching method to appropriately select the necessary information among the information of entity, motion, and textual representation. The final next word distribution, $P_t$, for the $t$-th step based on the switching method is defined as follows:

$$
P_t = \varsigma(W_v(r_t^1 P_t^e + r_t^2 P_t^m + r_t^3 P_t^T) + b_v) \in \mathbb{R}^{n_V},
\tag{11}
$$

and $W_v \in \mathbb{R}^{n_V \times n_P}$ and $b_v \in \mathbb{R}^{n_V}$ are learnable parameters. Each $r_t$ represents the importance of each representation when decoding the $t$-th word, and it is defined as follows:

$$
r_t = \varsigma(W_r h_t^D) \in \mathbb{R}^3, \qquad 0 \le r_t^k \le 1, \qquad \sum_{k=1}^{3} r_t^k = 1,
\tag{12}
$$

and $W_r \in \mathbb{R}^{3 \times n_{rw}}$ is learnable parameter.

According to this multi-representation switching, the decoder can separately consider the information of entities and motions in the video and the text information of text in the description. Through this, the proposed method can model entity, motion, and textual information separately.

A loss function for training is a negative log likelihood function for the target word $y^*$ and is defined as follows:

$$
\mathcal{L} = -\sum_{t=1}^{T} \log P_t[y_t^*].
\tag{13}
$$

## 4. Experiments

### 4.1. Dataset and Preprocess

For objective performance evaluation of the proposed method, we used the Microsoft Research Video Description (MSVD) dataset, which has been widely used in the video captioning problem. The MSVD dataset consists of 1970 YouTube clips and human-annotated descriptions in open domains such as cooking and movies. Each video has approximately 41 descriptions on average.

We tokenized the descriptions using the wordpunkt tokenizer from the Natural Language Toolkit and the descriptions were lowercased. Except for this space tokenization, no natural language processing technique was applied. This dataset has approximately 5.7 million tokens, and each description consists of approximately seven tokens. As in all other previous studies, we used 1200 videos for training, 100 videos for validation, and 670 videos for evaluation. We sampled a total of 20 frames twice per second from each video. We cropped each frame into a square shape and resized it to $224 \times 224$ pixels. To normalize the RGB values, we divided these values by 127.5 and subtracted them from 1. Sample training data from the MSVD dataset are shown in Table 2 with only five descriptions.

**Table 2.** Example of the training data.

| Video | Description |
|-------|-------------|
|  | a squirrel is eating a peanut in it's shell<br>a small animal is chewing on a nut<br>a chipmunk eats some food<br>a squirrel is eating a whole peanut<br>a hamster is eating a peanut |
|  | a person seasons eggs in a bowl<br>someone is mixing ingrediants<br>a person is cooking<br>a woman adds a mix of some sugar and salt to<br>a bowl containing two eggs<br>a man is talking about the recipe temaki sushi |
|  | a person peels on onion<br>a chef is peeling an onion<br>the lady peeled an onion<br>someone is peeling an onion<br>a person taking cover of onion |

From the descriptions of the first video in Table 2, we can observe that the same entity, a squirrel, was referred to by different words, such as "small," "animal," "chipmunk," or "hamster." Additionally, we can observe that the nut eaten by the squirrel was expressed by "peanut" or "food." Similar patterns can be observed in the descriptions of the second video. In addition, the word "ingrediants" is misspelled. The descriptions of the third video indicate an onion, but it can be confused with an orange if we do not pay attention to the video. Most of the descriptions were mainly composed in a progressive form.

For transfer learning, we used image datasets of the Microsoft Common Objects in Context dataset published in 2014 and the Flickr30k dataset. The images and descriptions were preprocessed in the same manner as the MSVD dataset.

*4.2. Experimental Environment*

We used the pretrained version of the ResNet50V2 network [27] with the ImageNet dataset. The dimensionalities for frame and word embeddings, $n_{ei}$, $n_{ew}$; the RNNs, $n_{ri}$, $n_{rw}$; the internal projection of attention mechanisms, $n_a$; and word logits, $n_P$, were set to 512. The maximum input, $I$, and output, $T$, lengths were set to 20 and 10, respectively. We used a leaky rectified linear unit [44] for the activation function. We used the Adam optimizer [45] for training with a learning rate of $5 \times 10^{-5}$, $\beta_1$ of 0.9, $\beta_2$ of 0.999, and $\epsilon$ of $1 \times 10^{-7}$. We only used words that appeared more than once in the training set of the MSVD dataset and the image datasets. In this setting, the vocabulary consisted of 21,992 tokens. The hyperparameters for the proposed method are listed in Table 3.

**Table 3.** Hyperparameters of the network structure of the proposed method.

| Hyperparameter | Value | Hyperparameter | Value |
|---|---|---|---|
| $I$ | 20 | $n_{ew}$ | 512 |
| $T$ | 10 | $n_{rw}$ | 512 |
| $n_{ei}$ | 512 | $n_a$ | 512 |
| $n_{kd}$ | [64, 256, 512, 1024, 2048, 1000] | $n_P$ | 512 |
| $n_{ri}$ | 512 | $n_V$ | 21,992 |

A beam search algorithm is a greedy tree search algorithm, and it limits the number of candidate groups for finding the optimal node to a beam size. To generate word sequences based on the trained method, many researchers have used the beam search algorithm. In

this case, the goal of the beam search algorithm was to find the combination of words with the highest likelihood in a greedy manner. We stored paths, whose termination token was generated separately, as candidates. We selected the word sequence with the highest average likelihood among the candidate group searched up to the maximum length and the previously stored candidate group. Because the length of the given description was short, no additional length penalty was applied. We used the beam search algorithm with a beam size of 5 to generate the descriptions.

Similar to other studies, we used the bilingual evaluation understudy (BLEU) [46], the metric for evaluation of translation with explicit ordering (METEOR) [47], and the consensus-based image description evaluation (CIDEr) [48] as performance measures. BLEU is a measure widely used for evaluating machine translation methods. It measures the n-gram precision based on automatically generated descriptions and the ground truth. We used BLEU-4 for evaluation. METEOR, likewise, is a measure widely used for evaluating machine translation methods, and it measures the n-gram precision in a semantic manner such as the form of stemming and synonym matching. CIDEr is a measure used to evaluate the image captioning method. It measures the n-gram F-score weighted by the term frequency–inverse term frequency. Higher scores on all these measures imply better performance.

To train the proposed method, we used a workstation equipped with an NVIDIA GeForce RTX 3090 graphics card to accelerate the training time. The proposed method was implemented using TensorFlow 2.3.

### 4.3. Single Model and Ensemble Model

We pretrained the proposed method for up to 10 epochs using the image datasets with a batch size of 32, and we trained the method for an additional 30,000 steps using the MSVD dataset with a batch size of 8. When training with the MSVD dataset, we stored the parameters at every 2000 steps. We denote this stored parameter as the single parameter $\theta_s$. In this work, 15 single parameters were stored, denoted by $\theta_s^{1,\ldots,15}$.

To further enhance the performance of the proposed method, we ensembled the single parameters following the previous research [49]. The ensemble method we used is an arithmetic average of the values of single parameters, $\overline{\theta}_e$, and a weighted average according to the METEOR scores of single parameters, $\widetilde{\theta}_e$. We chose the best five single parameters for the ensemble, and we also considered the combinations of the best two, the best three, and so on. These parameters are denoted by $\overline{\theta}_e^{1,\ldots,4}$ and $\widetilde{\theta}_e^{1,\ldots,4}$ from the best two to the best five, respectively. Among these 23 selected parameters, the parameter with the best validation performance was finally finetuned using validation data.

### 4.4. Quantitative Results

First, we validated the single parameters using the beam search algorithms and the performance measures, which were same as the test. The BLEU-4 and METEOR scores of the single parameters are shown in Figure 4.

In Figure 4, the highest point is marked by a blank circle, and its value is represented on the side of the circle. An index of optimal parameters is represented by a vertical red dashed line. As shown in Figure 4, with the highest BLEU-4 and METEOR scores, we chose the parameters, $\theta_s^3$, of 6000 ($3 \times 2000$) steps as the optimal single parameters, $\theta_s^*$, based on the validation set of the MSVD dataset. We chose the best five single parameters for the ensemble, and the parameters were $\theta_s^3$, $\theta_s^1$, $\theta_s^{10}$, $\theta_s^{15}$, and $\theta_s^{14}$ in the order of the METEOR score. The BLEU-4 and METEOR scores of the ensemble parameter are shown in Figure 5.

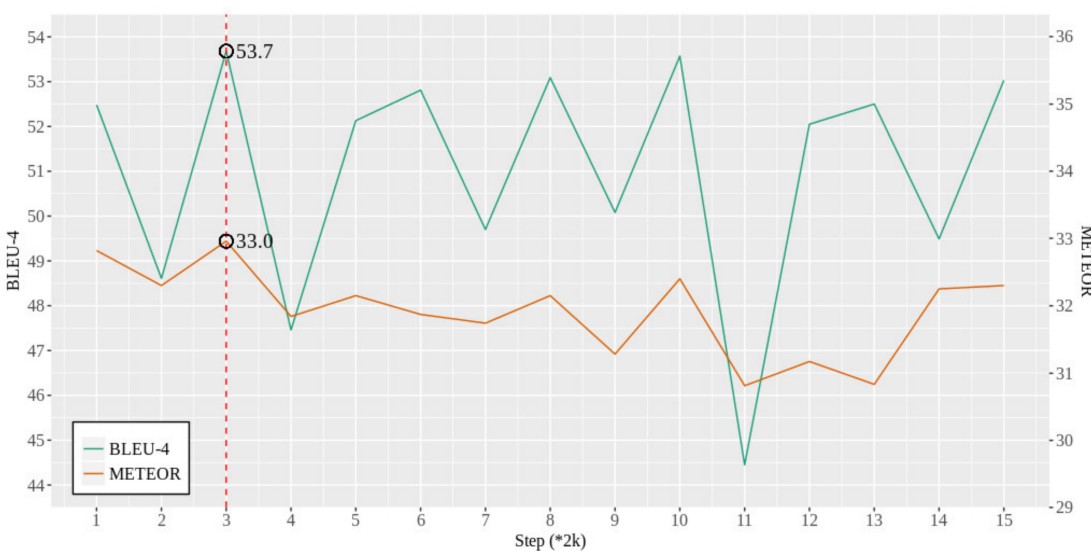

**Figure 4.** Validation BLEU-4 and METEOR scores of the single parameters.

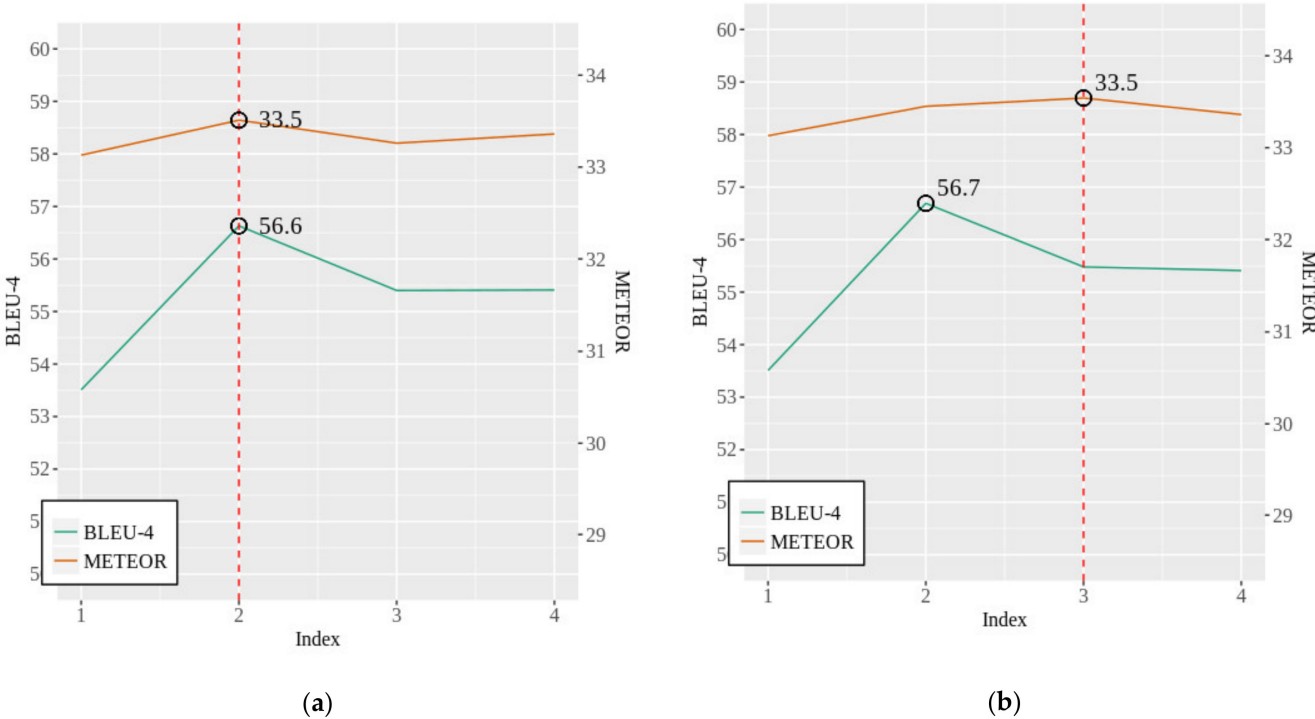

(**a**)  (**b**)

**Figure 5.** Validation BLEU-4 and METEOR scores. (**a**) Scores of the ensemble parameters of arithmetic average; (**b**) scores of the ensemble parameters of weighted average by METEOR score.

Through this validation, we finally chose the optimal parameters $\theta_s^* = \theta_s^3$, $\overline{\theta}_e^* = \overline{\theta}_e^2$, and $\widetilde{\theta}_e^* = \widetilde{\theta}_e^3$ with METEOR scores of 32.9, 33.51, and 33.54, respectively, for testing. The parameter with the highest METEOR score was $\widetilde{\theta}_e^* = \widetilde{\theta}_e^3$. We finetuned the parameter $\theta_e^*$ using the validation data of the MSVD dataset with a learning rate of 5e-6 for only one epoch, and the finetuned parameter is denoted by $\widetilde{\theta}_e^{*+}$. The proposed method with the optimal single parameters, $\theta_s^*$, is expressed as "Multi-Representation Switching with single parameter (MRS-s)" and the methods with the optimal ensemble parameters with arithmetic average, $\overline{\theta}_e^*$, and with weighted average, $\widetilde{\theta}_e^*$, are expressed as "MRS-ea" and "MRS-ew," respectively. Finally, the proposed method with the refined parameters, $\widetilde{\theta}_e^{*+}$, is expressed as "MRS-ew+."

The experimental results are shown in Table 4. The comparison methods are sorted in ascending order based on the METEOR score.

**Table 4.** Results of variant method and proposed method on Microsoft Research Video Description (MSVD) dataset.

| Models | BLEU-4 | METEOR | CIDEr |
|---|---|---|---|
| Meanpool [3] | 30.7 | 27.7 | - |
| SA-LSTM [5] | 41.9 | 29.6 | 51.7 |
| S2VT [2] | - | 29.8 | - |
| LSTM-GAN [12] | 42.9 | 30.4 | - |
| GRU-RCN [5] | 43.3 | 31.6 | 68.0 |
| BAE [7] | 42.5 | 32.4 | 63.5 |
| h-RNN [11] | 49.9 | 32.6 | 65.8 |
| PickNet [8] | 46.1 | 33.1 | 76.0 |
| STAT_V [6] | 52.0 | 33.3 | 73.8 |
| TSA-ED [9] | 51.7 | 34.0 | 74.9 |
| RecNet$_{local}$ [10] | 52.3 | 34.1 | 80.3 |
| MRS-s | 51.8 | 32.0 | 64.9 |
| MRS-ea | 52.4 | 32.9 | 74.8 |
| MRS-ew | 53.3 | 32.9 | 73.1 |
| MRS-ew+ | 54.3 | 34.0 | 80.3 |

As shown in Table 4, the proposed methods consistently showed good performance for the three measures. In the ensemble model, MRS-ea and MRS-ew, the BLEU-4 and METEOR scores increased slightly, but the CIDEr score increased significantly compared with the single model, MRS-s. In particular, the finetuned model, MRS-ew+, recorded scores that exceeded the performance of most existing video captioning methods. These scores were achieved in only 6000 training steps for MRS-s and 30,000 training steps for various MRS-e. Because the batch size was 8, one epoch consisted of approximately 6000 training steps, and it can be seen that these models were trained up to one epoch and five epochs, respectively.

We compared the proposed method with PickNet [8], STAT_V [6], TSA-ED [9], and RecNet [10], which are similar in performance and training method. PickNet was trained in three phases for up to 100 epochs in each phase. The first phase trains the encoder–decoder networks with a negative log likelihood loss for target words. The second phase trains the picking network based on reinforcement learning. The last phase jointly trains both networks. TSA-ED was trained for up to 30 epochs, or the training was stopped when the validation performance did not improve for 20 epochs. RecNet, similar to TSA-ED, was trained until the validation performance did not improve for 20 epochs. RecNet additionally used reconstruction loss for training. By contrast, the proposed method was only trained for up to five epochs.

The proposed method achieved BLEU-4, METEOR, and CIDEr scores that were 8.2 points, 0.9 points, and 4.3 points higher than PickNet, without reinforcement learning. Compared with TSA-ED, the proposed method recorded BLEU-4 and CIDEr scores that were 2.3 points and 5.4 points higher, and the METEOR score was tied. Compared with RecNet, the proposed method achieved BLEU and METEOR scores that were 2 points higher and 0.1 points lower, respectively, without additional loss, and the CIDEr score was tied.

STAT_V consists of a pretrained 2D CNN, C3D [40], and Faster R-CNN [41]. Owing to the characteristics of the MSVD dataset without additional coordinate information for videos, the R-CNN used in STAT_V cannot be finetuned in an end-to-end manner. This structure has a limitation in that the feature extraction phase and the description generation phase should be separated during training or testing. By contrast, the proposed method has the advantage that feature extraction and description generation phases are seamlessly connected so that each network can be smoothly finetuned. The proposed method achieved

BLEU-4, METEOR, and CIDEr scores that were 2.3 points, 0.7 points, and 6.5 points higher than STAT_V.

From this result, it can be confirmed that the proposed structure is effective for extracting information of a given video and description pair. The proposed method can be trained in a fully end-to-end manner, and we did not apply any preprocessing based on computer vision and natural language processing or any additional loss function.

### 4.5. Qualitative Results

To confirm the qualitative differences between the proposed method based on various parameters, we compared the generated descriptions for the test data. Examples of generating suitable descriptions are shown in Table 5.

**Table 5.** Examples of generating suitable descriptions.

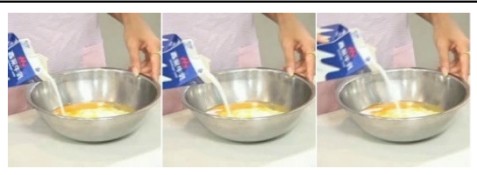 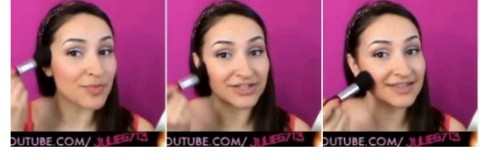

| | | |
|---|---|---|
| GT: | a woman adds milk to a bowl of cracked eggs | a woman is applying makeup to her face |
| MRS-s: | a person is stirring a bowl | a woman is doing her hair |
| MRS-ea: | a person is pouring a pot | a woman is singing |
| MRS-ew: | a person is cooking | a woman is singing |
| MRS-ew+: | a man is pouring ingredients into a bowl | a woman is applying makeup |

Generally, the proposed methods generated reasonable descriptions for the given videos. It seems that the higher the performance model, the more often rich expressions are used. Examples of matching a word with an incorrect entity and a correct entity are shown in Table 6.

**Table 6.** Examples of misunderstandings as similar entities.

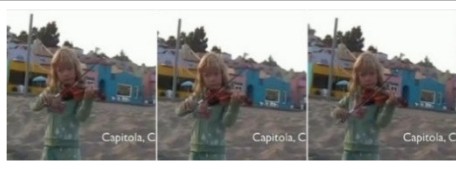 

| | | |
|---|---|---|
| GT: | a girl is playing a violin on the beach | a man playing a flute |
| MRS-s: | a girl is playing a flute | a man is crying |
| MRS-ea: | a girl is playing a flute | a man is playing |
| MRS-ew: | a girl is playing a flute | a man is playing |
| MRS-ew+: | a girl is playing a flute | a man is playing a flute |

The frames on the left represent a scene in which a violin is played; it seems that the violin was misrecognized as a flute because of the bow or face angles. The frames on the right show a scene where a man plays a flute, and we can observe that the face angles and instrument shape are similar in both the scenes. Other examples are listed in Table 7.

The frames on the left show a tomato, but it was not properly recognized. We believe that this is because of the green color of the tomato. The frames on the right show a gray rabbit playing with a pink rabbit doll. All the proposed methods recognized the rabbit as a dog, but MRS-ew+ was able to recognize the rabbit doll.

**Table 7.** Another example of misunderstanding similar entities.

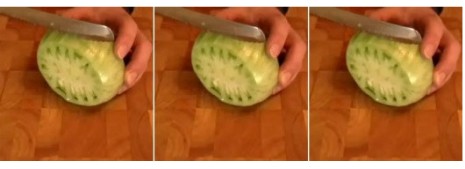 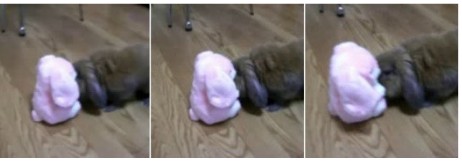

|  |  |  |
| --- | --- | --- |
| GT: | someone is slicing a green tomato with a knife | a gray rabbit is playing with a stuffed white rabbit |
| MRS-s: | a chef is cutting an onion | a puppy is playing |
| MRS-ea: | a woman is cutting an onion | a puppy is playing |
| MRS-ew: | a person is cutting an onion | a puppy is playing |
| MRS-ew+: | a man is cutting a onion | a puppy is playing with a rabbit |

## 5. Conclusions and Discussion

In this paper, we proposed a video captioning method based on multi-representation switching to model characteristics in a video and description pair. The proposed method consists of three components: one each for entity extraction, motion extraction, and textual feature extraction. The multi-representation switching method makes it possible for these three components to extract the important information for a given video and description pair as efficiently as possible.

Through the experiments using the MSVD dataset, the proposed method was learned according to the proposed intention, and achieved BLEU-4, METEOR, and CIDEr scores of 54.3, 34.0, and 80.3, respectively. These records exceed the performance of most existing methods with very few training steps and without any preprocessing based on computer vision, natural language processing, or any additional loss function. From the experimental results, we concluded the following for the research questions. A method that has a structure that embodies the characteristics of data can achieve good performance even with a compact feature extraction phase. In addition, this structure can be performed well even without additional regularization functions or training methods such as reinforcement learning. The simple structure of the proposed method can be used as a framework for various problems. In addition, it can be seen that the possibility of achieving higher performance is high when a complex feature extraction phase is conducted. Consequently, we can conclude that the proposed method has a high generality that can be extended to various domains in terms of sustainable computing.

A limitation of the proposed method is that it was often unable to distinguish between similar entities or motions. Research on structures that can explicitly deal with the correlation between merely indistinguishable entities or motions is left as future work.

**Author Contributions:** Conceptualization, H.K. and S.L.; Methodology, H.K.; Software, H.K.; Validation, H.K. and S.L.; Investigation, H.K.; Resources, S.L.; Writing—Original Draft Preparation, H.K.; Writing—Review and Editing, S.L.; Visualization, H.K.; Supervision, S.L.; Project Administration, S.L.; Funding Acquisition, S.L. All authors have read and agreed to the published version of the manuscript.

**Funding:** This research was supported by the Ministry of Science and ICT (MSIT), Korea, under the Information Technology Research Center (ITRC) support program (IITP-2020-2018-0-01419) supervised by the Institute for Information & Communications Technology Promotion (IITP).

**Institutional Review Board Statement:** Not applicable.

**Informed Consent Statement:** Not applicable.

**Data Availability Statement:** This dataset can be found at [MSVD] [50] [MSCOCO] https://cocodataset.org/#captions-2015 (accessed on 18 February 2021) [Flickr30k] [51].

**Conflicts of Interest:** The authors declare no conflict of interest.

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
