# Peer review of "A Video Captioning Method Based on Multi-Representation Switching for Sustainable Computing"

_sustainability, doi:10.3390/su13042250_

Round 1

Reviewer 1 Report

The article proposes a multi-representation switching method to extract the important information for a given video and description pair as efficiently as possible.

First, a good introduction to the research topic and the related works are given. In the further part, the proposed method is presented in detail and supported by examples and calculations. The fourth part presents the results of the application of the model using the Microsoft Research Video Description (MSVD) dataset.

The structure and explanations are clearly comprehensible. The proportions of the sections are balanced. The information is coherent and superfluous. The figure and tables are in the right places.

The authors’ contribution to the topic is important.

I have the following observations.

The research questions and hypotheses are missing.

The references must be formatted according to the journal requirements.

Line 350 is Qualiatative results and has to be “Qualitative….

Author Response

Thank you for your dedicate comments.

Following the comments, we revised the manuscript.

We added the paragraph for the research questions and hyhothesis in line 56-74.

We revised the format of references.

We revised the misspellings.

All changes are tracked by "Track changes" function of Word.

Sincerly.

Reviewer 2 Report

Authors propose the video captioning method based on multi-representation switching to model characteristics in the video and description pair. The proposed method consists of three parts, i.e., entity extraction, motion extraction, and textual feature extraction.

The article is expected to be relevant. However, authors mentioned about the sustainable computing in the last part and they have no discussion on it.

The article is assumed to be original, but authors are requested to expand the reference list and discuss the achievement of other researchers.

In the introduction authors could add the research method explanations. They presented an experiment. Although, the main posed question has been achieved, authors could explain more further development and usability of proposed solution.

Author Response

Thank you for your dedicate comments.

Following the comments, we revised the manuscript.

We added the paragraphs for the sustainable computing in line 56-74.

We expanded the references in the related works (from line 110), and we also added the disscussions the achievements in line 397-425.

We also added the paragraphs explaining further development and usability of the proposed method in line 462-469.

We revised the misspellings.

All changes are tracked by "Track changes" function of Word.

Sincerly.